# Choosing New Therapies for Gonorrhoea: We Need to Consider the Impact on the Pan-*Neisseria* Genome. A Viewpoint

**DOI:** 10.3390/antibiotics10050515

**Published:** 2021-05-01

**Authors:** Chris Kenyon, Jolein Laumen, Sheeba Manoharan-Basil

**Affiliations:** 1HIV/STI Unit, Department of Clinical Sciences, Institute of Tropical Medicine, 2000 Antwerp, Belgium; jlaumen@itg.be (J.L.); sbasil@itgg.e (S.M.-B.); 2Division of Infectious Diseases and HIV Medicine, University of Cape Town, Anzio Road, Observatory, Cape Town 7701, South Africa; 3STI Reference Center, Department of Clinical Sciences, Institute of Tropical Medicine, 2000 Antwerp, Belgium

**Keywords:** *N. gonorrrhoeae*, *Neisseria*, antimicrobial resistance, antimicrobial consumption, dual therapy

## Abstract

The development of new gonorrhoea treatment guidelines typically considers the resistance-inducing effect of the treatment only on *Neisseria gonorrhoeae*. Antimicrobial resistance in *N. gonorrhoeae* has, however, frequently first emerged in commensal *Neisseria* species and then been passed on to *N. gonorrhoeae* via transformation. This creates the rationale for considering the effect of gonococcal therapies on resistance in commensal *Neisseria*. We illustrate the benefits of this pan-*Neisseria* strategy by evaluating three contemporary treatment options for *N. gonorrhoeae*—ceftriaxone plus azithromycin, monotherapy with ceftriaxone and zoliflodacin.

## 1. Introduction

*Neisseria gonorrhoeae* (NG) causes the disease gonorrhoea, which is an important cause of urethritis, cervicitis and proctitis. NG has developed resistance to every class of antimicrobial used to treat it [1]. Antimicrobial resistance (AMR) is not, however, inevitable. There are large differences in the prevalence of AMR between different countries [2]. In some populations, such as the Northern Territories of Australia, there is so little AMR that oral penicillin can still be used to treat NG [2,3]. In other countries such as China, high levels of resistance to azithromycin (18.6%) and reduced susceptibility to ceftriaxone (12.2%) compromise even these last-line, single-dose therapies [4]. In general, the prevalence of AMR to a range of antibiotics has increased in most countries. Thus, the World Health Organization’s Global Gonococcal Antimicrobial Surveillance Programme (GASP) has found continuing high-level resistance to penicillin, tetracycline and ciprofloxacin around the world. It has also found increasing resistance to azithromycin and the emergence of decreased susceptibility and resistance to cephalosporins such as cefixime and ceftriaxone [5]. Cephalosporin and macrolide resistance in NG is of particular concern. In the United States (USA), the percentage of isolates with reduced susceptibility (MIC ≥ 2.0 μg/mL) increased from 0.6% in 2013 to 4.6% in 2018 [6]. Likewise, in the 24 countries participating in the European Gonococcal Antimicrobial Surveillance Programme (Euro-GASP), azithromycin resistance increased from 5.3% in 2011 to 13.3% in 2018 (MIC > 1.0 μg/mL) [6]. Whilst there have only been a limited number of isolates with ceftriaxone reported from Europe and the USA, the high prevalence of reduced susceptibility to ceftriaxone reported from China and elsewhere is a major concern [4].

The key determinant of AMR is antimicrobial exposure [7]. As the antimicrobials used to treat gonorrhoea are an important driver of this antimicrobial exposure, treatment guidelines are based in part on minimising the probability that they will induce AMR in NG [6,8]. Treatment of NG will, however, also select for AMR in commensal *Neisseria,* which are an important component of the oropharyngeal microbiome. These commensal *Neisseria* can then transfer this resistance to NG [9]. Since horizontal gene transfer (HGT) has played a crucial role in the genesis of AMR in NG, we make a case for explicitly incorporating treatment effects on commensal *Neisseria* when evaluating prospective therapies for NG. We describe this as the pan-*Neisseria* perspective.

## 2. Contemporary Monospecies Approach

Treatment guidelines for gonorrhoea consider the risk that new treatments could induce AMR but typically only consider this effect on NG. For example, recent United States CDC and European IUSTI NG treatment guidelines both consider resistance induction in their recommended treatment guidelines. Neither includes commensal *Neisseria* in this evaluation [6,8]. In the case of the CDC guidelines, “concerns regarding potential harm to the microbiome” are listed as one of the reasons for changing recommended treatment from ceftriaxone plus azithromycin (dual therapy) to ceftriaxone (monotherapy) [8]. The IUSTI guidelines focus exclusively on resistance induction in NG and argue that because concurrent resistance to ceftriaxone and azithromycin is extremely rare, dual therapy has likely played a role in decreasing the prevalence of resistance to ceftriaxone and cefixime [6]. The authors further note that there have been examples of therapy failure with monotherapy, but none with dual therapy (Figure 1). These are among the arguments used to recommend dual therapy as the preferred therapy for NG in the IUSTI guidelines. Thus, whilst the CDC explicitly recommends monotherapy, IUSTI recommends dual therapy.

It is important to note that approaches that only consider resistance induction in the target organism (mono-species approaches) may be more likely to favour poly-therapy than approaches that consider resistance induction in multiple organisms (Table 1). This is evident if we consider the example of therapy for *Mycobacterium tuberculosis*, wherein standard, four-drug treatment dramatically reduces the probability of the emergence of AMR compared to single drug therapy [10]. Four drug therapy reduces the probability of AMR emerging. This relates to the fact that a single mycobacterium is considerably less likely to be resistant to four drugs than one drug. Resistance is caused by chromosomal mutations, which occur at a frequency of approximately 10^−6^ mycobacterial replications [10]. The probability of a single mycobacterium being resistant to one antimicrobial is thus around 10^−6^, but this jumps to 10^−12^ if two antimicrobials are considered [11]. This explains why, when single drug therapy with streptomycin was introduced, it was rapidly followed by resistance to streptomycin, which was not the case with quadri-therapy [10]. A key conclusion is that for organisms like *M. tuberculosis*, where AMR is acquired via chromosomal mutations, monotherapy selects for, and quadri-therapy prevents, the emergence of AMR. The same principles have been shown to apply to other organisms wherein HGT is not prominent such as other species of mycobacterium and *Plasmodium falciparum* [12]. The situation is, however, more complicated in bacteria like *Neisseria* spp., wherein HGT plays an important role in the genesis of AMR [13].

## 3. Pan-Neisseria Approach

The antimicrobials used to treat NG select for AMR in both pathogenic and commensal *Neisseria*. This effect has been shown for cephalosporins, fluoroquinolones and macrolides [14,15,16]. The genetic mutations responsible for this AMR can be readily taken up from these commensals by NG via transformation [17].

This process of transformation has played a critical role in the emergence of NG resistance to both cephalosporins and macrolides [17,18,19]. Mosaic versions of *penA* are a crucial determinant of gonococcal cephalosporin resistance [17]. These mosaic *penAs* emerged via multiple independent acquisitions of sections of the *penA* gene from a number of commensal *Neisseria,* including *N. cinerea, N. mucosa, N. subflava* and *N. lactamica* [17,20,21]. Likewise, transformation from commensal *Neisseria* has played an important role in the emergence of resistance to azithromycin in NG. As an example, NG has taken up portions of the genes coding for the mtrCDE efflux pump from commensal *Neisseria* [18,19]. This has enabled it to more effectively pump out azithromycin from the intracellular compartment and thereby become less susceptible to azithromycin.

The available evidence suggests that high levels of antimicrobial consumption are the major determinant of the increase in commensal AMR during the past few decades [14,15,22]. There are a number of different types of antimicrobial consumption that can result in commensal AMR (Figure 2), and it is useful to consider this selection pressure at both the individual and population levels [13]. As illustrated in Figure 1 and Figure 3, dual therapy at the level of an infected individual would very likely eradicate NG but not commensal *Neisseria*. In part, this is related to differences in resilience.

NG at an individual level has little or no resilience. Effective antimicrobial therapy leads to eradication, with close to zero probability of recurrence without reinfection (Figure 4). Commensal *Neisseria*, in contrast, are highly resilient. Numerous studies have confirmed that close to 100% of all humans are colonised by specific, commensal *Neisseria* species in various oropharyngeal niches [23,24,25,26,27]. For example, although their distribution and abundance varied, *N. mucosa*, *N. subflava* and *N. flavacens* were all found to be present in the oral microbiomes of all persons included in the Human Microbiome Project [28]. This proportion remains remarkably constant despite various antimicrobial challenges [15,25]. For example, the proportion of individuals colonised by commensal *Neisseria* is unchanged 14 days post-dual ceftriaxone/azithromycin therapy [15,29]. We do not understand the factors underpinning this remarkable resilience of commensal *Neisseria*, but there is increasing evidence that they are better considered as synergists that provide a number of vital functions to their human hosts [23,24,25,26,27]. Whatever the mechanism, the available evidence suggests that commensal *Neisseria* are able to withstand broad spectrum antimicrobials and antiseptic mouthwashes that are able to eradicate other bacteria. We hypothesise that the universal presence of commensal *Neisseria*, combined with their resilience, increases the probability of AMR emerging in high-antimicrobial-exposure settings in two ways. Firstly, the higher prevalence increases the probability of antimicrobial exposure and hence bystander AMR selection [30]. This is evident if we consider that a species that is present in 100% of a population will be much more likely to be exposed to antimicrobials used for any indication than a species whose prevalence is 1%. The second factor is what we have termed the ‘force of resistance’ effect (Figure 3). This refers to the extent to which antimicrobial consumption depresses a species prevalence below its equilibrium prevalence in a particular population. Consider a species that has an equilibrium prevalence of 100% and a high propensity to return to 100% prevalence post-antimicrobial exposure. If antimicrobial therapy drives prevalence down to 20%, this will likely exert a greater selection pressure for the emergence of AMR than if the prevalence was driven down to 95% [9,13]. The ‘force of resistance’ effect is likely largely determined by the extent to which the prevalence of a bacterial species is reduced below its equilibrium prevalence at a population level or below its normal abundance at an individual level. In the case of NG, the ‘force of resistance’ only operates at a population level, wherein the equilibrium prevalence of a population is determined by its sexual network connectivity [9]. If this prevalence is reduced with widespread antimicrobial consumption, then the greater the decline, the greater the selection pressure for AMR to emerge (which would enable the NG to return to its equilibrium prevalence; Figure 3 and [9]). In the case of commensal *Neisseria*, this ‘force of resistance’ operates at both individual and population levels (Figure 3). One of the population-level mechanisms for AMR to spread within commensals is illustrated in Figure 4, where high antimicrobial consumption leads to relative extinction of susceptible strains and hence an increase in the transmission of resistant strains between individuals via activities such as kissing, which has been shown to be a mechanism for the spread of commensal *Neisseria* [31,32].

If this line of reasoning is correct, then it would follow that high antimicrobial consumption would be manifested in earlier and higher AMR in commensal *Neisseria* than NG. Furthermore, there should be a correlation between antimicrobial consumption and antimicrobial susceptibility in commensal *Neisseria*. Whilst more evidence is required in this regard, the available evidence is generally supportive. The commensal *Neisseria* MICs of azithromycin and ceftriaxone are significantly higher in HIV preexposure prophylaxis (PrEP) clients in Belgium than those from the general Belgian population, whose consumption of these antimicrobials is considerably lower (unpublished data). In men who have sex with men (MSM) taking PrEP in Belgium, we have found that the median MIC for the most prevalent commensal, *N. subflava,* for example, increased from 1 in historical samples (1980–2000) to 176 mg/L in 2019 [15]. When we explored the potential drivers of this increase in resistance, we found that the azithromycin used for dual therapy of NG resulted in a macrolide consumption of 9.5 to 12 (DID) defined daily doses/1000 inhabitants/day [33,34]. This is approximately 4 to 7 times higher than the thresholds estimated to induce macrolide resistance in a range of bacterial species [35]. In vitro experiments demonstrated that NG was able to acquire macrolide resistance from DNA extracts from these highly macrolide-resistant *N. subflavas* via transformation [36]. The results for ceftriaxone were similar—there has been a large rightward shift in the MICs of commensal *Neisseria* in MSM taking PrEP and exposed to high levels of cephalosporins. These commensals were found to contain a number of mutations in *penA*, *porB*, and *mtrCDE* that can be taken up by NG and result in macrolide and cephalosporin resistance [29].

There is also considerable supportive evidence for the resilience of commensal *Neisseria*. As already noted, one study found that oropharyngeal commensal *Neisseria* were as abundant 14 days post-therapy with ceftriaxone/azithromycin as pre-treatment [29]. This study also found that the median azithromycin and ceftriaxone MICs of all isolated *Neisseria* were higher in the post-treatment isolates [15,29]. Likewise, a study of commensal *Neisseria* in MSM in Vietnam found high cephalosporin MICs that were strongly correlated with recent receipt of a cephalosporin [14]. These findings suggest that whilst dual therapy is highly efficacious in eradicating NG and other susceptible *Neisseria*, it selects for macrolide- and cephalosporin-resistant commensal *Neisseria* (and other bacteria). In the case of oral streptococci, macrolide consumption has been shown to result in an increase in the proportion of isolates with macrolide resistance—an effect that persists for over 6 months [37]. Azithromycin’s effect on the abundance of resistance-associated genes in the gastrointestinal tract may persist for four years [38,39].

## 4. Differences Stemming from Monospecies and Pan-*Neisseria* Approaches

These findings suggest important difference between the monospecies and pan-*Neisseria* approaches as regarding therapeutic choices for NG (Figure 1; Table 1). The monospecies approach only considers the risk of resistance induction in NG. It is thus more likely to favour dual therapy, as this combination may minimise the risk of treatment failure, which could result in resistance [6]. The pan-*Neisseria* approach, however, incorporates the effect of proposed therapies on AMR in commensal *Neisseria*. It is likely to be more circumspect about dual therapy that includes azithromycin, especially considering that its long half-life means the drug is present for at least 2 to 4 weeks post-treatment in decreasing concentrations [40]. This azithromycin tail poses little risk of induction of AMR from a monospecies perspective (as the NG has been eradicated) [40] but a large risk for inducing AMR from a pan-*Neisseria* perspective, as commensal *Neisseria* are in the process of reconstituting their populations during this period [9,30].

As already noted, dual therapy in conjunction with screening for NG in MSM PrEP cohorts results in very high consumption levels of macrolides-9.5 to 12 DID [33,34]. Whilst we do not know what the thresholds for the emergence of AMR are in commensal *Neisseria*, the fact that this consumption is 4- to 7-fold higher than the thresholds for inducing AMR in other organisms and the very high azithromycin MICs of commensal *Neisseria* spp. isolated from these PrEP cohorts suggests that the macrolide resistance threshold is considerably lower than 12 DID.

To ascertain variables such as these resistance inducing thresholds, the pan-*Neisseria* approach would advocate introducing surveillance of commensal *Neisseria* antimicrobial susceptibility in key populations (Table 1). The other variables that could be obtained from such a process and be used to model the relationship between antimicrobial consumption and AMR in NG and commensals are detailed in Figure 2. This figure describes the six main pathways linking antimicrobial consumption and AMR in NG/commensal Neisseria. The relative importance of these pathways may vary by type of antimicrobial considered. Thus, macrolides such as azithromycin appear to be particularly adept at inducing NG resistance indirectly via the commensal pathway [18]. This has important implications for the use of macrolides for a range of infections/indications. Macrolides are frequently used for treatment of infections caused by organisms such as *Chlamydia trachomatis, Mycoplasma genitalium* and *Streptococcus pneumoniae*, where alternative agents are available [41,42,43]. They are also used long-term to prevent exacerbations of chronic obstructive pulmonary disease. The pan-*Neisseria* approach would recommend that bystander selection be considered when developing treatment guidelines for these infections.

An important criticism of the pan-*Neisseria* approach is that whilst HGT was important in the genesis of macrolide and cephalosporin resistance, these events were infrequent, and the majority of spread of this resistance has been clonal [1,18,19]. Whilst there is merit in this argument, there is also evidence that episodes of HGT in *penA*, *mtrCDE, gyrA* and other loci have been frequent in the pathogenic *Neisseria* [16,17,20,21]. Whilst there are no longitudinal studies evaluating the incidence of HGT in NG, a study that followed up with a cohort for 6 months found high incidence of HGT between *N. meningitidis* and *N. lactamica*. Evidence of HGT between these two *Neisseria* species during the follow-up period was detected in 15 loci in the two individuals that were co-colonised by both bacteria at baseline [44]. Finally, the clonal spread of these mosaic genes within NG is itself strongly influenced by antimicrobial consumption [45,46]. By including consideration of AMR in commensal *Neisseria*, one may be able to detect the emergence of AMR at an earlier and reversible stage and use this information to reduce consumption of the relevant antimicrobial in the target population.

## 5. Application of the Pan-*Neisseria* Approach to Novel Treatments

We end by illustrating how the pan-*Neisseria* approach could be used to evaluate the promising novel anti-NG agent, zoliflodacin, currently in phase 3 trials (Figure 2) [47]. Following the monospecies approach, current in vitro evaluations of this agent have been limited to assessing how easily zoliflodacin resistance can be induced in NG [48,49,50]. Likewise, assessments of the prevalence of pre-existing, resistance-conferring *gyrB* mutations have been considered in NG and not commensal *Neisseria* [47]. A pan-*Neisseria* approach would complement these activities by including a panel of commensal *Neisseria* in these evaluations. If zoliflodacin resistance could be induced in commensal *Neisseria* relatively easily in vitro, then one could assess how easily this resistance could be transformed into NG in subsequent experiments.

It may be objected that we have no evidence of HGT ever having taken place in *gyrB* in *N. gonorrhoeae*. Whilst this is true, we have good evidence that HGT in *gyrA* played an important role in the genesis of fluoroquinolone resistance in *N. meningitidis*. A study in Shanghai found that 99.3% of commensal *Neisseria* and 67.7% of *N. meningitidis* isolates were resistant to fluoroquinolones, and that HGT from commensals was responsible for fluoroquinolone resistance in over half the *N. meningitidis* isolates [16]. The most plausible reason for this extremely high prevalence of fluoroquinolone resistance in *Neisseria* species is the high consumption levels of fluoroquinolones in the general population [16]. In silico analyses of gonococcal and commensal *gyrA* from around the world have revealed that HGT has played a similarly important role in the genesis of fluoroquinolone resistance in *N. gonorrhoeae* (unpublished data).

These findings provide the motivation to include surveillance of commensal *Neisseria* susceptibility to zoliflodacin in clinical trials and implementation projects of zoliflodacin. This could complement conventional AMR surveillance that monitors zoliflodacin MICs in NG. It would be particularly important to do in core groups with high exposure to anti-gonococcal therapies, such as PrEP cohorts [51]. Such a surveillance system could act as an early warning system for the emergence of zoliflodacin AMR in NG. One might find that zoliflodacin resistance was easily transformed from commensals into NG, and zoliflodacin AMR emerged faster in commensals than NG. If this was the case, then one could decide to switch/cycle the recommended NG treatment from zoliflodacin to another agent once predefined resistance thresholds in commensal *Neisseria* were crossed.

The pan-*Neisseria* approach builds on the insights of authors such as Bacquero et al., who have noted the utility of conceptualising epidemics of AMR as occurring simultaneously in multiple species [52]. They noted that epidemics of extended spectrum beta lactamase (ESBL) resistance in Gram negative bacteria occurred in a range of species rather than a single species. Excess use of cephalosporins resulted in outbreaks of ESBL producers in multiple species, partly driven by these species sharing the resistance-conferring enzymes with one another. In a similar vein, other authors have found correlations between fluoroquinolone resistance in various Gram negatives and *N. gonorrhoeae* at a country level [53]. In all cases, a key underlying driver of AMR is excessive antimicrobial consumption. This creates the rationale to select early warning species that can be used in surveillance programs to provide an alert when antimicrobial consumption is becoming excessive. Commensal *Neisseria* may be useful for this purpose, particularly in populations with a high STI incidence [14,15].

The pan-*Neisseria* approach raises a large number of important research questions. How does AMR in commensals vary by time and population (including core groups with different intensities of antimicrobial exposure)? Are there thresholds for the emergence of AMR in commensals? Is AMR reversible? How effective would antimicrobial cycling be to prevent AMR? What is the probability of HGT between *Neisseria* spp. in vivo? Do antimicrobial induced changes to commensal *Neisseria* populations have other adverse health effects?

Ultimately, all knowledge is underpinned by theory. An optimal theory of the determinants of gonococcal AMR should provide an accurate portrayal of all the important determinants in a way that illustrates the interrelationships and the relative importance of the various determinants and facilitates proportionate and effective responses [54]. If the theory conceals certain determinants, then it should be replaced by a theory which does not do this [54,55]. From the evidence we have reviewed, we conclude that a pan-*Neisseria* theoretical approach should be preferred to a monospecies approach, as it provides a more complete understanding of the genesis and spread of AMR in NG.

## 6. Conclusions

The development of new gonorrhoea treatment guidelines typically considers the resistance-inducing effect of the treatment only on *Neisseria gonorrhoeae*. Antimicrobial resistance in *N. gonorrhoeae* has, however, frequently first emerged in commensal *Neisseria* species and then been passed on to *N. gonorrhoeae* via transfer of the relevant resistance genes. This creates the rationale for considering the effect of gonococcal therapies on resistance in commensal *Neisseria* as well as on *N. gonorrhoeae*.

## Figures and Tables

**Figure 1 antibiotics-10-00515-f001:**
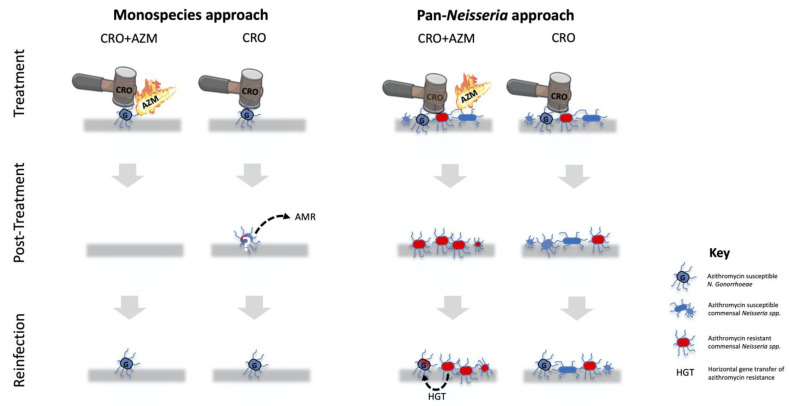
A schematic illustration of select differences between the monospecies and pan-*Neisseria* approaches to evaluating the risk of inducing resistance of proposed antimicrobial therapy for *N. gonorrhoeae* (NG/G). The monospecies approach only considers the effects of the treatment on NG. It thus favours dual therapy with ceftriaxone (CRO) and azithromycin (AZM), as this combination minimises the risk of treatment failure that could result in resistance. In the pan-*Neisseria* approach, monotherapy is favoured, as this is highly efficacious at eradicating NG and does this without selecting for widespread resistance (red bacteria) to macrolides in commensal *Neisseria* 14 days post-therapy (second panel). The pan-*Neisseria*, but not the monospecies approach, is cognisant of the risk of the genes conferring macrolide resistance in commensal *Neisseria* being passed on to a NG reinfection via horizontal gene transfer (HGT; third panel). The rationale for only representing CRO and CRO/AZM as treatment options is that these are the predominant treatments currently recommended by the United States CDC and European IUSTI guidelines.

**Figure 2 antibiotics-10-00515-f002:**
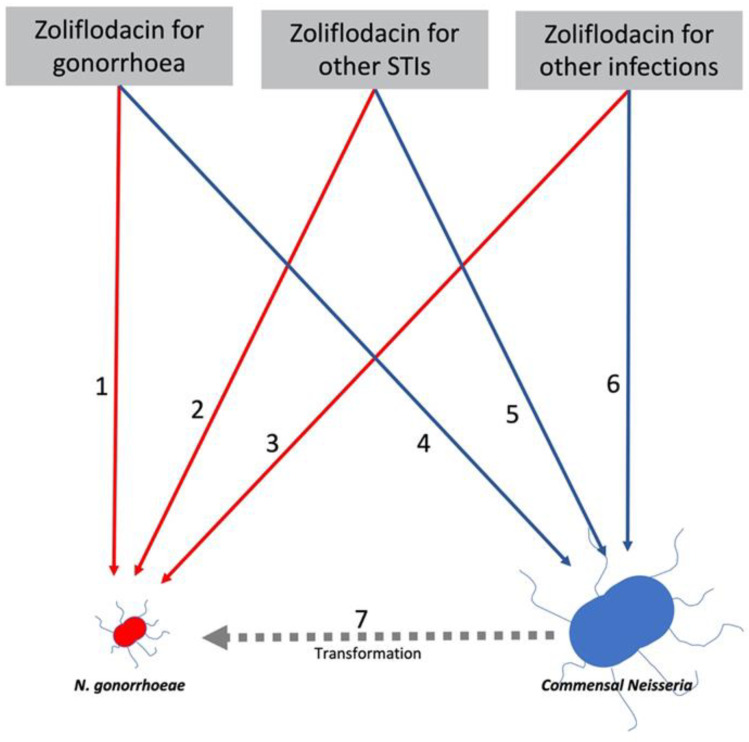
Model of relationship between zoliflodacin consumption for 3 indications and induction of zoliflodacin resistance in *N. gonorrhoeae* (NG: red) and commensal *Neisseria* (blue). Zoliflodacin is a promising, novel anti-gonococcal therapy that also shows promise for a range of other infections. Zoliflodacin (ZF) usage could select for AMR directly in NG via 1. ZF used to treat NG, 2. ZF used to treat other STIs and 3. ZF used to treat other infections. It could also select for AMR in commensal *Neisseria* via 4. ZF used to treat NG, 5. ZF used to treat other STIs or 6. ZF used to treat other infections. Each of these could select for ZF resistance in commensal *Neisseria*, which could then be transferred to NG via transformation (7). The probability of ZF inducing AMR in NG (1,2,3) and commensal species (4,5,6) could be determined via in vitro experiments. These experiments could also estimate the efficiency of transformation given the co-occurrence of NG and a specific commensal *Neisseria* (7). The commensal *Neisseria* in the figure is bigger than NG to reflect the orders of magnitude of the higher prevalence of commensal *Neisseria*.

**Figure 3 antibiotics-10-00515-f003:**
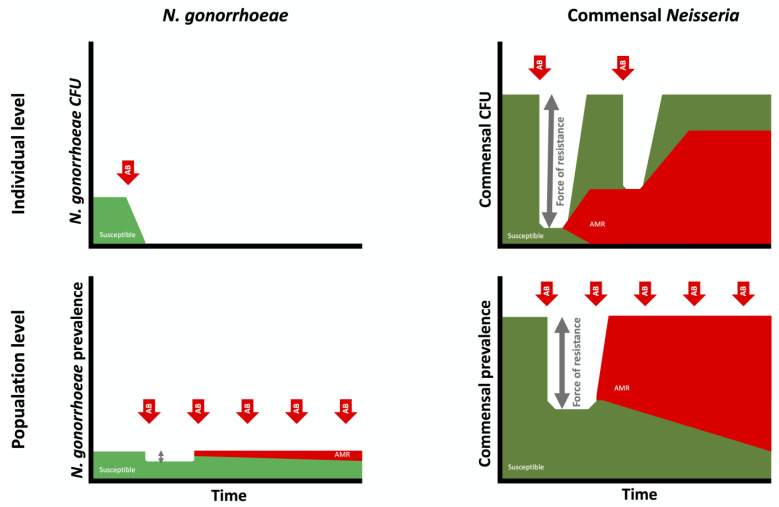
Schematic illustration of the importance of considering the selection of antimicrobial resistance in *N. gonorrhoeae* (**left**) and commensal *Neisseria* (**right**) at both the individual (**top**) and population levels (**bottom**). Whereas dual therapy with ceftriaxone/azithromycin (AB; red arrows) eradicates *N. gonorrhoeae* in an infected individual, it only reduces the abundance of commensal *Neisseria*. The resilience of commensal *Neisseria* results in their returning to baseline abundance in a process that selects for resistant isolates. The greater the decline in abundance of commensal abundance, the greater the ‘force of resistance’ (grey arrows). At a population level, high exposure to ceftriaxone/azithromycin will push the prevalence of both *N. gonorrhoeae* and commensal *Neisseria* species below the equilibrium prevalences (100% for commensals and 0.1–10% for *N. gonorrhoeae*, depending on sexual network connectivity). This will select for resistance to these agents in both *N. gonorrhoeae* and commensal *Neisseria*. (CFU: colony forming units).

**Figure 4 antibiotics-10-00515-f004:**
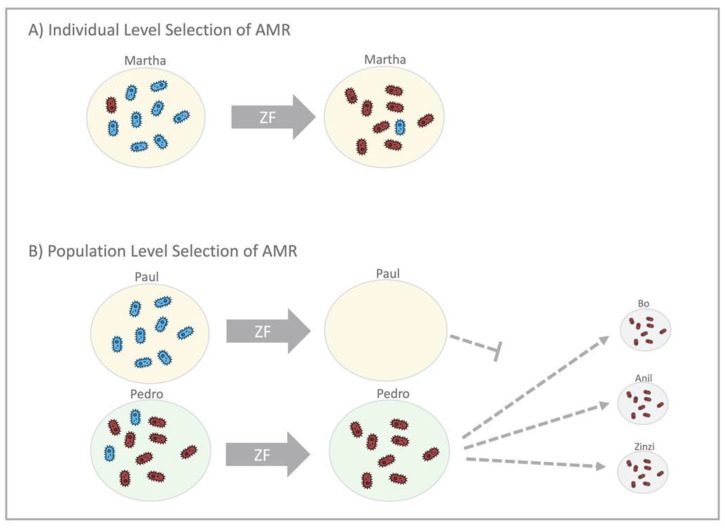
An illustration of individual- and population-level mechanisms that select for antimicrobial resistance (AMR) in commensal *Neisseria*, using the example of zoliflodacin (ZF). (**A**) In the individual level selection scenario (top), Martha takes ZF, and this eradicates the ZF-susceptible *Neisseria* (blue bacteria), leaving Martha with predominantly ZF-resistant *Neisseria* (red bacteria) post-treatment. (**B**) Recent studies have found that commensal *Neisseria* can be transmitted by kissing. Population-level selection of AMR by widespread use of ZF (bottom) works by eradicating the susceptible *Neisseria* from Paul and Pedro, leaving only resistant *Neisseria* to be transmitted by both of them to others.

**Table 1 antibiotics-10-00515-t001:** Select differences between the monospecies and pan-Neisseria approaches to selecting new *N. gonorrhoeae* (NG) therapies.

	Monospecies Approach	Pan-*Neisseria* Approach
Conceptual framework	Monospecies conception: NG is a pathogen, and its control requires optimisation of seek and destroy activities.	Ecological conception: Commensal *Neisseria* are important constituents of a healthy microbiome and can be a source of AMR for NG and *N. meningitidis*. Excessive seek and destroy activities could induce AMR in commensals, which could be transferred to NG.
Approach to dual therapy (ceftriaxone + azithromycin) vs. monotherapy (ceftriaxone) for NG	Treatment with dual therapy is favoured, as this is more likely to eradicate NG than monotherapy.	Dual therapy for NG is more likely to have a negative effect on commensals (composition and macrolide resistance) and hence, monotherapy may be preferable.
AMR Surveillance	Surveillance in samples of NG is sufficient, e.g., Euro GASP, GRASP methodologies	Surveillance should be done in both NG and commensal *Neisseria* in core groups, e.g., culture/MIC of commensal *Neisseria* from throat swabs of 30 PrEP clients per centre once a year.

## Data Availability

Not applicable.

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
