# Peer review of "Choosing New Therapies for Gonorrhoea: We Need to Consider the Impact on the Pan-Neisseria Genome. A Viewpoint"

_antibiotics, 2021, doi:10.3390/antibiotics10050515_

Round 1

Reviewer 1 Report

The authors present an interesting viewpoint concerning the current guidelines for Neisseria gonorrhoeae treatment. 

General comments

Overall, I found the manuscript interesting but there was some information lacking. 

Moreover, the English should be revised, there are a lot of wordy phrases, which impairs the reading. I point some but they are only a few examples, the text should all be revised.

The figures' placement is odd and impairs the reading. Most are placed at the end of the manuscript, why not the first time they are mentioned? Figure 4 is mentioned before Figure 3, why not switching the numbering?

At the end of the manuscript, the authors kept some sections with the journals' instructions. Please revise their content and if they are needed. 

Finally, when submitting a manuscript for revision, the document should contain line numbers to assist the reviewer in his work. The line numbers are usually embedded in the journal's template.

Specific comments 

  • The third sentence in chapter 1 is not very clear or in accordance with the remaining text. the authors start by saying that NG has developed resistance against all antibiotics available, and then follows to "low prevalence of resistance", what should the readers retain?
  • Numbers should be mentioned at the beginning of the manuscript. What is the real prevalence of NG resistance to the current antibiotics? How is this impacting the treatments? The recommendation for dual therapy is theory-based, or the resistance of this species is big enough to harm patients' treatment with a monotherapy? There are numbers regarding treatment failure with mono vs dual therapy? What is the percentage of success associated with a mono-therapy? Does it eradicate the disease? Or the mono-therapy can lead to the re-activation of resistant strains that ultimately would need much stronger antibiotics? 
  • Chapter 2 - Maybe I miss something. The CDC recommends changing the dual to monotherapy, but the IUSTI recommends the opposite? If yes, more explanations should be given.
  • Figure 1 - What happens if Neisseria is resistant to ceftriaxone? Why only focusing on azithromycin? What is so special or different about this antibiotic that justifies this distinction? The authors should include this information at the beginning of the manuscript.
  • Right after figure 1's legend - what the Mtb approach show is that a multi-drug regimen is highly efficient in escaping AMR (the first part of the first sentence but not the second part of the sentence), revise these sentences.
  • The last sentence of page 2 should be revised.
  • Keep in mind that, the rationale behind the quadri-therapy against Mtb is not only based on resistance but also on the different states the bacteria can present inside the host, the different locations the bacteria can have, and the accessibility of the dugs to it. Moreover, as is the case of ethambutol, its main function is to give access to other drugs to the interior of mycobacteria. My main point is, the reasoning behind mycobacteria treatment (Mtb or not) goes much further than just resistance.
  • First sentence of chapter 3 - Also? Are you not talking about this in the last pages? It seems like a new theme is being introduced.
  • Chapter 3 "In a similar vein," it's the type of sentence that is too wordy and should be revised.
  • Page 4, "Secondly, if a species..." the sentence is big and wordy, the reader is lost in the middle.
  • What does MSM stand for?
  • Page 5: "In vitro experiments were able to... was able to..." revise.
  • Chapter 4, third sentence - numbers?
  • Chapter 4: not "includes consideration of" but considers.
  • Finally, the authors should make some comment (a focused one) on the use of azithromycin for a series of infections and how would this impact this pan-Neisseria approach.

Author Response

Please see the attachment for the new manuscript

Reviewer 1

The authors present an interesting viewpoint concerning the current guidelines for Neisseria gonorrhoeae treatment. 

General comments

Overall, I found the manuscript interesting but there was some information lacking. 

Moreover, the English should be revised, there are a lot of wordy phrases, which impairs the reading. I point some but they are only a few examples, the text should all be revised.

Reply:

The language has been extensively revised as suggested to improve readability. The specific edits suggested below have been made.

The figures' placement is odd and impairs the reading. Most are placed at the end of the manuscript, why not the first time they are mentioned? Figure 4 is mentioned before Figure 3, why not switching the numbering?

Reply:

Thank you for pointing this out. This layout has been corrected.

At the end of the manuscript, the authors kept some sections with the journals' instructions. Please revise their content and if they are needed. 

Reply:

These sections have been removed.

Finally, when submitting a manuscript for revision, the document should contain line numbers to assist the reviewer in his work. The line numbers are usually embedded in the journal's template.

Reply:

Line numbers have been added.

Specific comments 

  • The third sentence in chapter 1 is not very clear or in accordance with the remaining text. the authors start by saying that NG has developed resistance against all antibiotics available, and then follows to "low prevalence of resistance", what should the readers retain?

Reply:

A new sentence has been added to clarify this point (L29-30)

  • Numbers should be mentioned at the beginning of the manuscript. What is the real prevalence of NG resistance to the current antibiotics? How is this impacting the treatments? The recommendation for dual therapy is theory-based, or the resistance of this species is big enough to harm patients' treatment with a monotherapy? There are numbers regarding treatment failure with mono vs dual therapy? What is the percentage of success associated with a mono-therapy? Does it eradicate the disease? Or the mono-therapy can lead to the re-activation of resistant strains that ultimately would need much stronger antibiotics? 

Reply:

A new paragraph has been added to the manuscript to address these issues (L39-51)

  • Chapter 2 - Maybe I miss something. The CDC recommends changing the dual to monotherapy, but the IUSTI recommends the opposite? If yes, more explanations should be given.

Reply:

A new sentence has been added to provide further clarification (L 72-73)

  • Figure 1 - What happens if Neisseria is resistant to ceftriaxone? Why only focusing on azithromycin? What is so special or different about this antibiotic that justifies this distinction? The authors should include this information at the beginning of the manuscript.

Reply:

The justification for choosing ceftriaxone and ceftriaxone plus azithromycin is that these are the predominant treatments currently recommended by major guidelines around the world. This information has been added to the manuscript (L87-89). Similar arguments could be made with different combinations of antibiotics.

  • Right after figure 1's legend - what the Mtb approach show is that a multi-drug regimen is highly efficient in escaping AMR (the first part of the first sentence but not the second part of the sentence), revise these sentences.

Reply:

This has been revised (L92)

  • The last sentence of page 2 should be revised.

Reply:

This has been revised (L97-99)

  • Keep in mind that, the rationale behind the quadri-therapy against Mtb is not only based on resistance but also on the different states the bacteria can present inside the host, the different locations the bacteria can have, and the accessibility of the dugs to it. Moreover, as is the case of ethambutol, its main function is to give access to other drugs to the interior of mycobacteria. My main point is, the reasoning behind mycobacteria treatment (Mtb or not) goes much further than just resistance.

Reply:

This is an important point. We could add a section along these lines if the reviewer so desires? We have elected not to do this thus far, as in our opinion, it is not crucial information for the point we are trying to make. The point we are making is that one of the reasons for the using quadritherapy for MTB is that this will prevent resistance emerging.

  • First sentence of chapter 3 - Also? Are you not talking about this in the last pages? It seems like a new theme is being introduced.

Reply.

The word ‘also’ has been deleted.

  • Chapter 3 "In a similar vein," it's the type of sentence that is too wordy and should be revised.

Reply:

This section has been rewritten (L135-139)

  • Page 4, "Secondly, if a species..." the sentence is big and wordy, the reader is lost in the middle.

Reply:

This section has been rewritten (L198-201)

  • What does MSM stand for?

Reply:

MSM refers to men who have sex with men and has been defined (L263)

  • Page 5: "In vitro experiments were able to... was able to..." revise.

Reply:

“In vitro experiments” is plural and thus it would be grammatically incorrect to use the third person singular (was). The third person plural (were) is the correct choice.

  • Chapter 4, third sentence - numbers?

Reply:

No randomized controlled trials have been performed comparing the mono- and dual-therapy approaches. As such it is not possible to provide numerical estimates.

  • Chapter 4: not "includes consideration of" but considers.

Reply:

Thank you for this useful suggested edit, which has been made.

  • Finally, the authors should make some comment (a focused one) on the use of azithromycin for a series of infections and how would this impact this pan-Neisseria approach.

Reply:

A new paragraph has been outlined to address this request (L338-346)

Reviewer 2 Report

  1. Briefly summarise the content of the manuscript; The manuscript concerns the new approach to the antimicrobial resistance of Neisseria gonorrhoeae (gonococcus), which is a causative agent of gonorrhea. This obligatory human pathogen is the only bacterial etiological agent of sexually transmitted disease, which, due to its widespread antibiotic resistance, has been classified as a "priority pathogen" by the World Health Organization in 2017. The Authors consider the impact of pan-Neisseria genome on antimicrobial resistance of Neisseria gonorrhoeae and on effective anti-Neisseria therapies in the context of contemporary treatment options for N. gonorrhoeae.
    2. Illustrate what are, in your opinion, the manuscript’s strengths and weaknesses (this is an essential step, because the Editor will consider the reasoning behind your recommendation); The manuscript’s strengths: The authors presented the monoscpecies and Pan-Neisseria approach to antibiotic resistance of N. gonorrhoeae, for example they indicate differences between monospecies and pan-Neisseria approach to evaluation of the risk of introducing resistance of putative anti-Neisseria therapy. The Authors provide the molecular basis of antibiotic resistance of Neisseria. They point out that the commensal Neisseria can be antibiotic resistant and thus it can be a source of antibiotic resistance of N. gonorrhoeae. The manuscript’s weaknesses
    Some parts of the manuscript make it difficult to read as a whole, e.g the section on Mycobacterium and the link between information on Neisseria and Mycobacterium is missing. This part should be rewritten.
    3. Provide a point-by-point list of your recommendations for the improvement of the manuscript (in case that you have any recommendations);
    1. As I mentioned above, the link between information on Neisseria and Mycobacterium is missing. This should be completed.
    2. The information if such approach was already applied to another antibiotic resistant microorganism should be also added.

Author Response

Please see the attachment for the new version of the paper

Reviewer 2

  1. Briefly summarise the content of the manuscript; The manuscript concerns the new approach to the antimicrobial resistance of Neisseria gonorrhoeae (gonococcus), which is a causative agent of gonorrhea. This obligatory human pathogen is the only bacterial etiological agent of sexually transmitted disease, which, due to its widespread antibiotic resistance, has been classified as a "priority pathogen" by the World Health Organization in 2017. The Authors consider the impact of pan-Neisseria genome on antimicrobial resistance of Neisseria gonorrhoeae and on effective anti-Neisseria therapies in the context of contemporary treatment options for N. gonorrhoeae.
    2. Illustrate what are, in your opinion, the manuscript’s strengths and weaknesses (this is an essential step, because the Editor will consider the reasoning behind your recommendation); The manuscript’s strengths: The authors presented the monoscpecies and Pan-Neisseria approach to antibiotic resistance of N. gonorrhoeae, for example they indicate differences between monospecies and pan-Neisseria approach to evaluation of the risk of introducing resistance of putative anti-Neisseria therapy. The Authors provide the molecular basis of antibiotic resistance of Neisseria. They point out that the commensal Neisseria can be antibiotic resistant and thus it can be a source of antibiotic resistance of N. gonorrhoeae. The manuscript’s weaknesses
    Some parts of the manuscript make it difficult to read as a whole, e.g the section on Mycobacterium and the link between information on Neisseria and Mycobacterium is missing. This part should be rewritten.
    3. Provide a point-by-point list of your recommendations for the improvement of the manuscript (in case that you have any recommendations);
    1. As I mentioned above, the link between information on Neisseria and Mycobacterium is missing. This should be completed.

Reply:

This has been section has been expanded (see reply to reviewer 1)

  1. The information if such approach was already applied to another antibiotic resistant microorganism should be also added.

Reply:

A new section to this effect has been added (L 408-411)

Round 2

Reviewer 1 Report

Dear Authors, 

although many of the suggested changes were made, the writing style is still poor, I strongly recommend that you revise your text thoroughly. 

Some examples (please, do not change just these, but revise the entire manuscript):

Lines 45-46 - isolated and isolates in the same sentence, the authors can not find synonyms for one of these words?

Line 90 - The word probability is written 3 times in the same (long) sentence. 

Line 92 - "...than the probability of being resistant to...

Line 245 - I am quite aware of the singular and plural rules for verbs conjugation. My point was that, in the same sentence, the authors write "were able to" followed by "was able to" - the sentence has much room for improvement. 

Line 310 - Infections caused by - the authors list the microorganisms' names, not the infections/diseases' names.

Line 376 - "as occurring as allodemics"? The English is not very clear, and the allodemics concept should be introduced, for all the readers who are not familiarised with it.

Moreover, the absence of numbers regarding the real antimicrobial resistance of Neisseria to the available antimicrobials is a huge gap in this manuscript. The point of this work is to propose a new strategie for therapy reccomendation, then, the current global situation should be discussed more clearly. The authors only add two sentences about it, it is not enough. 

Lines 27 and 28 - What countries? How is this difference? How could this impact giving a monotherapy instead of a dual-therapy? It took me only 2 minutes to check the eppidemological data on the WHO website to see that resistance to ceftriaxone is rapidly increasing in Europe. 

Reviewer 2 Report

The manuscript has been improved. However, few additionally corrections are required.  

For example:

  • colloquial sentences should be avoided, e.g. “The story is”
  • there is the mess with citation in the manuscript; in the one place, the authors use the square brackets, and in the other one round brackets.
  • source the information about macrolides used for treatment of infections caused by Chlamydia trachomatis, Mycoplasma pneumonia and Streptococcus pneumoniae. Lines 308-316.
